# Mechanical Properties of Five Esthetic Ceramic Materials Used for Monolithic Restorations: A Comparative In Vitro Study

**Saleh N. Almohammed \*** , **Belal Alshorman and Layla A. Abu-Naba'a**

Department of Prosthodontics, Faculty of Dentistry, Jordan University of Science and Technology,
Irbid 22110, Jordan
* Correspondence: snalmohammed@just.edu.jo

**Abstract:** Monolithic zirconia and hybrid ceramic restorations have been widely used in the last decade for both anterior and posterior dental restorations. However, their use lacks sufficient scientific evidence in most cases, as the expeditious manufacturing of these versatile ceramic materials exceeds the limits of in vitro and/or in vivo validation. This study aimed to evaluate and compare the mechanical properties (flexural strength, fracture toughness, Vickers hardness, and brittleness index) of three CAD-CAM monolithic multilayer zirconia ceramics (GNX—Ceramill Zolid® Gen-X, ZCP—IPS e.max® ZirCAD, and UPC—Upcera® Esthetic Explore Prime) and one CAD-CAM monolithic multilayer polymer-infiltrated hybrid ceramic (ENM—Vita® Enamic) with a CAD-CAM monolithic lithium disilicate ceramic as a control (EMX —IPS e.max® CAD). A total of 160 discs (GNX = 32, ZCP = 32, UPC = 32, ENM = 32, and EMX = 32) were cut, polished, and fully sintered (except for the ENM). Half of the samples for each group were subjected to hydrothermal aging. Descriptive analysis and ANOVA tests were used to compare the groups. The zirconia groups showed significantly higher mechanical properties than the EMX group for both the non-aged and aged samples ($p < 0.05$). The ENM group showed the lowest brittleness index, while EMX showed the highest. The mechanical properties of monolithic multilayer zirconia ceramics were generally better than those of monolithic multilayer polymer-infiltrated hybrid ceramic and lithium disilicate ceramic. All groups showed, to some extent, a change in their mechanical properties after aging, with the ENM being the most affected.

**Keywords:** aging; brittleness; flexural strength; hardness; lithium disilicate; mechanical properties; monolithic; polymer-infiltrated ceramic; toughness; zirconia



## 1. Introduction

The introduction of CAD-CAM technology enabled the use of high-strength ceramic materials in clinical settings, such as alumina and zirconia. Every year, new materials enter the market, claiming to be esthetically pleasing, monolithic, and CAD-CAM compatible. The expanding esthetic expectations, as well as the capabilities for in-office manufacturing of restorations via CAD-CAM designing, milling, and sintering equipment, are the primary driving forces behind this vast market [1].

In dentistry today, three types of zirconia systems are used: yttria-stabilized zirconia polycrystals (Y-SZP), magnesia-partially-stabilized zirconia (Mg-PSZ), and ceria-stabilized zirconia/alumina nanocomposite (Ce-TZP/A). Each type is basically the crystalline oxide of the element zirconium at room temperature and is composed of white zirconia with different melting points [2]. The quantity of yttria in the material influences the strength value of the zirconia. In general, zirconia with a higher yttria content will have lower strength but higher translucency. Using 3 mol% yttria provides a zirconia material with high strength, often in the 900–1200 MPa range. This is due to the high tetragonal phase composition (85%–90%), which results in outstanding mechanical characteristics. These needed to be veneered to mask their opaque color. Chipping of the feldspathic veneering

layer that masks the ceramic core opacity became a major issue, thus a shift towards translucent zirconia ceramics, primarily 4Y-SZP and 5Y-SZP, have been attempted to address this issue. This allows the material to be employed monolithically, that is, a dental restoration is produced entirely from the same material, thus having the same unity without the use of a veneering layer [3].

Raising the yttria content to 5 mol% resulted in a strength drop, typically in the 500–900 MPa range. This concentration produced a more transparent material with around 50% cubic phase content. Zirconia materials with 6 and 8 mol% yttria generate more transparent materials at the sacrifice of mechanical strength. These materials' strengths are generally in the 300–600 MPa range [4]. It is crucial to note that additional parameters, such as sintering conditions, grain size, and the presence of impurities, influence zirconia strength values. Also, depending on the testing procedure, the particular strength levels might vary. To maximize the required qualities, it is critical to carefully analyze the intended usage of the zirconia material and pick the optimum yttria concentration.

Quite apart from their benefits, high-strength ceramic materials have been limited in their application in locations with high occlusal stresses due to their very brittle nature, a tendency to produce wear on opposite natural teeth, and chipping of the veneering ceramic coating [1]. Another constraint that may have an impact on its clinical performance is "aging," also known as "low-temperature deterioration," which is the spontaneous change of the metastable tetragonal zirconia structure into the more stable monoclinic phase. This transition takes place in a humid atmosphere at low temperatures (65–300 °C). Degradation can also happen when zirconia is exposed to different oral circumstances, including exposure to aqueous conditions, temperature fluctuations, dietary acidity, and fatigue loading during chewing cycles [5]. This transition may initially benefit the zirconia structure by increasing the compressive layer on the surface and therefore boosting its mechanical characteristics. More aging, on the other hand, would be detrimental to the characteristics of the zirconia since it causes the propagation of macro and microfractures, in addition to grain pull-out and outer layer roughening [6].

By integrating hybrid resins or polymers with ceramics, more monolithic and esthetically attractive ceramic material alternatives have now been made available [7]. These hybrid materials combine high structural strength with more compliant and esthetic resin polymers to provide the best of both worlds. RBC CAD-CAM blocks can be either resin nanoceramics or polymer-infiltrated ceramics (PIC). Because RBCs are easier to grind, less costly, and easier to repair than ceramic-based blocks, they seem to be gaining favor and growing fast in the market. Additionally, they feature elastic moduli similar to real dentition. Nanoceramics are composed of approximately 86% ceramic and 14% resin particles by weight. The inclusion of a large fraction of nanoparticles in the resin matrix increases material resistance to breaking and abrasion. Regardless of its high ceramic filler, this material is best suited for inlays, onlays, and veneers rather than crowns [8].

Mechanical testing of dental materials is critical, even if they pass the ISO standards [9], as different commercial brands claim to meet them. While ISO standards set guidelines for the testing of dental materials, they do not guarantee that the materials will perform optimally in clinical situations. The performance of dental materials can be influenced by various factors, including the manufacturing process, storage conditions, and handling during clinical use. Therefore, it is essential to cross-check and verify the mechanical properties of dental materials through rigorous testing to ensure that they are suitable for their intended use. Mechanical testing can provide valuable information on the strength, durability, and wear resistance of dental materials, which are critical factors for their long-term clinical success. Hence, the importance of mechanical testing cannot be overstated, and it should be an integral part of the quality control process for dental materials [10].

The aim of this study was to report and compare the mechanical properties of five esthetic ceramic materials used as monolithic CAD-CAM restorations, as well as to determine how hydrothermal aging affected these properties (flexural strength, fracture toughness,

Vickers hardness, and brittleness index). This test comes after these new commercial materials' optical characteristics have already been investigated elsewhere [11].

The null hypothesis to be tested is that, according to ISO standards, (1) there is no difference in the mechanical qualities (flexural strength, fracture toughness, Vickers hardness, and brittleness index) of zirconia ceramics, hybrid ceramics, and lithium disilicate ceramics and (2) the mechanical characteristics of the ceramic materials examined are unaffected by artificial aging.

## 2. Materials and Methods

CAD-CAM monolithic materials are the subject of this research. Three were multilayer zirconia ceramics (GNX-Ceramill Zolid® Gen-X, ZCP-IPS e.max® ZirCAD, and UPC-Upcera® Esthetic Explore Prime) and one CAD-CAM monolithic multilayer polymer-infiltrated hybrid ceramic (ENM-Vita® Enamic). These were compared to a standard esthetic CAD-CAM monolithic material (EMX-IPS e.max® CAD LT). All samples were pre-shaded and labeled (A2) and were drawn from the same lot number of blocks or discs, as shown in Table 1. The materials chosen here were selected as they range mechanically from high-zirconia crystals ($ZrO_2$ + $HfO_2$ + $Y_2O_3$ > 99.0%) to zero-percent zirconia filler. Some of these compositions have been clinically used and mechanically tested for over a decade, even before being customized for CAD-CAM applications.

To establish the sample size, a power analysis was done using G*Power statistical software (G*Power Ver. 3.0.10, Franz Faul, Universität Kiel, Kiel, Germany). Samples for each category were chosen with the following set criteria: power: 0.8, $\alpha$: 0.05, and effect size: 0.5; for assessing mechanical qualities in each group, a sample size of 32 was used.

A total of 160 samples were prepared, which were then separated into five groups ($n$ = 32). Each group represented one of the five materials evaluated, which were separated as follows: 18 samples for the flexural strength test (9 with and 9 without aging), 14 for the hardness and fracture toughness tests (7 samples with and 7 without aging).

The GNX ($n$ = 32), ZCP ($n$ = 32), and UPC ($n$ = 32) samples with the appropriate dimensions for each test were dry-milled from partly sintered zirconia blanks using a CAM machine (Ceramill® motion 2, Amann Girrbach, Pforzheim, Germany). The sample cutting dimensions were calculated with the knowledge that monolithic zirconia shrinks by 20–25% during dense sintering, as stated by the manufacturer.

A water-cooled milling CAM machine (Ceramill® motion 2, Amann Girrbach, Pforzheim, Germany) was used to harvest ENM ($n$ = 32) and EMX ($n$ = 32) samples from their respective blocks (as recommended). The cutting measurements of the EMX samples were calculated while considering a 0.2–0.3% shrinkage throughout crystallization. ENM requires no extra heat treatment. The final dimensions of the sample discs measured with a digital caliper (Guanglu, Gullin, China) were within 10mm diameter and 10.02mm thickness. A total of 30 samples were created from a single supplied disc unit for the GNX, ZCP, and UPC, and 2 from each of the ENM and EMX blocks.

In this study, disc-shaped samples were chosen, and during sample shaping or polishing, cooling with water or no cooling was conducted according to the manufacturers' specifications. The discs were then polished with 600, 800, and 1000 grit silicon carbide (SiC) papers without water in a grinding device (echo LAB POLI-1X/250, Devco S.r.l, Paderno Dugnano, Milan, Italy) for zirconia samples but with water for hybrid ceramic and lithium disilicate samples, as per the manufacturers' suggestions. Before any thermal processing, each polishing round was accomplished by one operator for 60 s at 300 rpm. This step was done before sintering because samples prepared from pre-sintered Y-TZP blocks deformed the surface layer with micro-cracks appearing at the surface. The final sintering process partially heals the micro-cracks and eliminates voids and flaws. Polishing after this would produce more surface scratches and heat changes [12]. So, to standardize the samples' surface roughness, polishing removed surface residual stresses and fissures caused by intrinsic material imperfections or production methods; however, it is not used in clinical practice (Figure 1).

**Table 1.** Materials used in this study.

| Material as Described by the Company | Subtype Numbers and Company Details | Trade Name and Specific Subtype Tested | Abbreviation | Basic chemical Structure (Chemical Composition, (wt%) |
|---|---|---|---|---|
| Multilayer highly translucent monolithic zirconia (4 layers) | Ceramill Zolid has 7 subtypes, Amann Girrbach AG, Germany | Ceramill Zolid® Gen-X | GNX | 4Y-TZP:<br>$ZrO_2 + HfO_2 + Y_2O_3 \geq 99.0\%$<br>$Y_2O_3$ 6–7%<br>$HfO_2 \leq 5\%$<br>$Al_2O_3 \leq 0.5\%$<br>Other oxides $\leq 1\%$ |
| Natural esthetics and high-strength multilayer zirconia (3 layers) | IPS ZirCAD has 5 subtypes, Ivoclar Vivadent, Schaan, Liechtenstein | IPS e.max® ZirCAD Prime | ZCP | 3Y-TZP and 5Y-TZP:<br>$ZrO_2$ 88–95.5%<br>$Y_2O_3$ 4.5–7%<br>$HfO_2 < 5\%$<br>$AL_2O_3 < 1\%$<br>Other oxides < 1.5% |
| Monolithic multilayer zirconia (5 Layers) | Upcera has 11 subtypes, Upcera, China | Upcera® Esthetic Explore Prime | UPC | 4Y-TZP and 5Y-TZP:<br>$ZrO_2 + HfO_2$ 86.3–94.2%<br>$Fe_2O_3 < 0.5\%$<br>$Y_2O_3$ 5.8–9.7%<br>$Er_2O_3 < 2\%$<br>$Al_2O_3 < 0.5\%$<br>Other oxides < 0.5% |
| Polymer infiltrated hybrid (Single layer) | Vita Enamic has 3 subtypes, Zahnfabrik H. Rauter GmbH, Germany | Vita® Enamic 2M2 T | ENM | 86% by weight (75% by volume) ceramic network and 14% by weight (25% by volume) polymerized methacrylate polymer network; UDMA and TEGDMA [$SiO_2$ (58–63%), $Al_2O_3$ (20–23%), $Na_2O$ (9–11%), $K_2O$ (4–6%), $B_2O_3$ (0.5–2%), CaO (<1%) and $TiO_2$ (<1%)]. |
| Highly esthetic lithium disilicate (Single layer) | IPS e.max CAD has 4 subtypes, Ivoclar Vivadent, Schaan, Liechtenstein | IPS e.max® CAD LT | EMX | $SiO_2$ 57–80 %<br>$Li_2O$ 11–19%<br>$K_2O$ 0–13%<br>$P_2O_5$ 0–11%<br>$ZrO_2$ 0–8%<br>ZnO 0–8%<br>$Al_2O_3$ 0–5%<br>MgO 0–5%<br>Coloring oxides 0–8% |

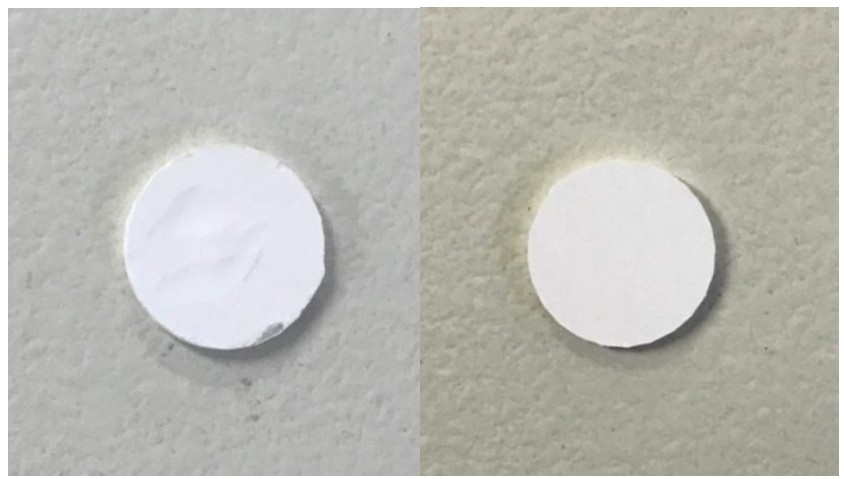

**Figure 1.** Unpolished and polished zirconia sample before sintering.

The zirconia specimens GNX, ZCP, and UPC then were completely sintered in a sintering furnace (Ceramill Therm; Amann Girrbach, Pforzheim, Germany), whereas the lithium disilicate EMX samples were crystallized in a furnace (Programat EP5010; Ivoclar-Vivadent, Liechtenstein) based on the manufacturer's instructions. There was no firing or crystallization heat treatment for the ENM hybrid ceramic, as indicated by the manufacturer, as it is fully cured. The details are presented in Table 2 and Figure 2.

**Table 2.** Sintering/Crystallization parameters used for tested materials.

| Group | Heating Rate and Eventual Heating Steps | Final Temperature (°C) | Holding Time (min) | Cooling Rate up | Furnace Brand |
|---|---|---|---|---|---|
| GNX | 8 °C/min | 1450 °C | 120 | 20 °C/min | Ceramill Therm (Amann Girrbach) |
| ZCP | 10 °C/min until 900 °C is attained; after holding for 30 min, use a heating rate of 3.3 °C/min until 1500 °C | 1500 °C | 120 | 10 °C/min from 1500 °C to 900 °C, then 8 °C/min from 900 °C to 300 °C | Ceramill Therm (Amann Girrbach) |
| UPC | 10 °C/min until 300 °C, then 17.5 °C/min until 1000 °C, and 4 °C/min until 1530 °C | 1530 °C | 120 | 12.2 °C/min | Ceramill Therm (Amann Girrbach) |
| EMX | 60 °C/min until 770 °C is attained, hold for 5 min, then 30 °C/min until 850 °C | 850 °C | 10 | 20 °C/min | Programat EP5010 (Ivoclar Vivadent) |

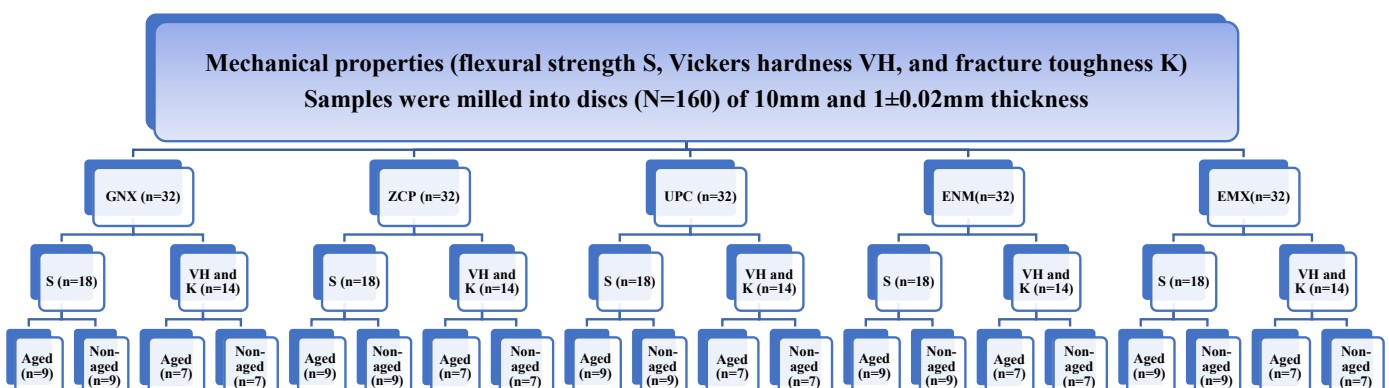

**Figure 2.** Distribution of samples in the study groups.

### 2.1. Aging Procedure

A 5-h (15–20-year age) hydrothermal cycle in a steam autoclave (Euronda B type, Italy) at 134 °C and 2 bar (200 KPa) was performed on half of each group's 16 samples ($n$ = 8). This methodology was selected because the one-hour autoclave aging process at 134 °C and 2 bar (200 KPa) corresponds to three to four years of clinical usage and promotes thorough tetragonal-monoclinic phase transformation with an estimated 55–80% monoclinic phase content [13]. Separators were utilized during this process to separate specimens packed in sterilizing sealing packets.

### 2.2. Biaxial Flexure Test (S)

A computer-controlled mechanical universal testing apparatus (Jinan Testing Machine, WDW-20, China) and a test setup in line with ISO standard 6872 were used to measure the load at fracture (Newton) of the various ceramic disc groups. After sintering, the 90 samples (45 with and 45 without aging) were held for 24 h in distilled water at 37 °C. The disc support portion in the lower component of the testing equipment was constructed from three 3.2 mm steel balls arranged at a 120° angle to each other, generating a 10 mm circle in

the disc housing's bottom (ISO Standard 6872, 2015), as shown in Figure 3. The upper part was equipped with a flat circular tungsten piston (r = 0.7 mm), which was employed by the universal testing machine to deliver an escalating load of 1 mm/min until catastrophic collapse occurred.

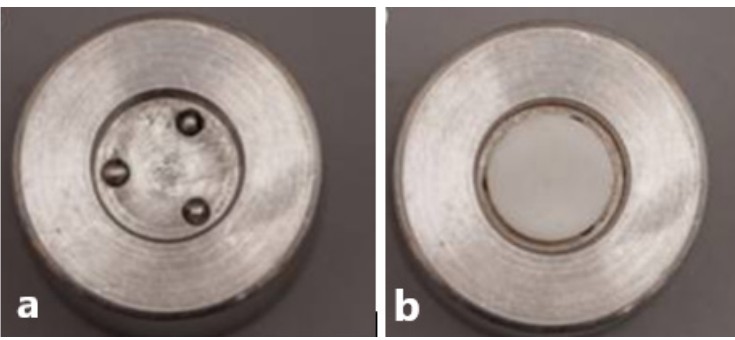

**Figure 3.** (**a**) Metal housing device with three balls to support the discs. (**b**) Specimen positioned in contact with three balls.

The load at the fracture point was measured and the biaxial flexural strength (piston-on-three-ball test) at each sample was computed using equation [14]:

$$S = -0.2387\ P\ (X - Y)/d^2,$$

where S is the maximum center tensile stress (MPa) (the flexural strength at fracture) and P is the total load causing fracture (N).

$$X = (1 + \nu)\ln(r2/r3)^2 + [(1 - \nu)/2](r2/r3)^2 \text{ and}$$

$$Y = (1 + \nu)[1 + \ln(r1/r3)^2] + (1 - \nu)(r1/r3)^2,$$

in which ν is Poisson's ratio,
  r1 is the radius of the support circle (5 mm),
  r2 is the radius of the loaded area (0.7 mm),
  r3 is the radius of the specimen (5 mm), and
  d is the specimen thickness at the origin of the fracture (1 mm) [15].

### 2.3. Vickers Hardness (VH)

After sintering, samples (n = 70) were kept in distilled water at 37 °C for 24 h (35 with aging and 35 without aging). After storage, the samples were evaluated for hardness with a microhardness indenter (Micromet 5101, Buehler, Lake Bluff, IL, USA). Three indents with a 120-degree angle in between were made near the center of each specimen at least 2.5 mm from the disc's center.

A Vickers indenter with a weight of 1kg and a dwell period of 15 s was used in accordance with ASTM C1327 standards. The primary diagonals of the Vickers indent (d1 and d2) were measured using an optical microscope and hardness was calculated using the following formula [16]:

$$VH = 1855 \times Load/(d1 \times d2),$$

where d1 and d2 are the major diagonals of the Vickers indent under an optical micrometer, as shown in Figure 4.

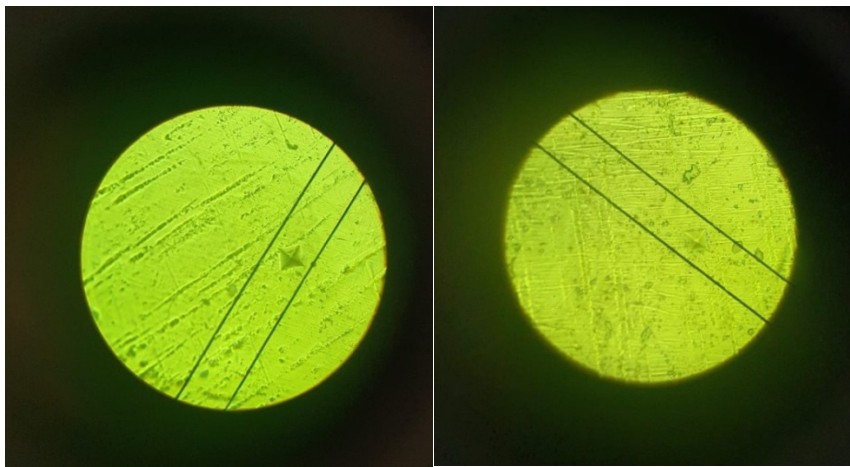

**Figure 4.** The tested sample under the microscope lens.

*2.4. Fracture Toughness*

According to Anstis et al. [17], and as shown in Figures 5 and 6, the crack length, C (measured from the middle of the indent), needs to be at least equal to or larger than the diagonal length (2a). The conventional Vickers loads of the hardness apparatus were first examined to determine the optimal load for fulfilling the C/a 2 criteria. As a result, this was established via experimentation and error.

Fracture toughness was estimated using this equation [18]:

$$K = 0.016 \times (E/H)^{1/2} \times P/C^{3/2}$$
$$H = P/(2a)^2$$
$$C/a \geq 2,$$

where K is the fracture toughness of the material (MPa.m$^{1/2}$),

H is the hardness,
E is the elastic modulus, as shown in Table 3,
P is the load applied (N), as shown in Table 3,
a is the indent half diagonal (μm), and
C is the crack length measured from the center of the indent (μm).

**Table 3.** Elastic moduli and Poisson's ratio of the ceramics used in the study [19].

| Material | Elastic Moduli | Poisson's Ratio |
|---|---|---|
| Polymer-infiltrated ceramic network | 30 GPa | 0.28 |
| Lithium disilicate | 95 GPa | 0.25 |
| Yttria-stabilized zirconia | 200 GPa | 0.31 |

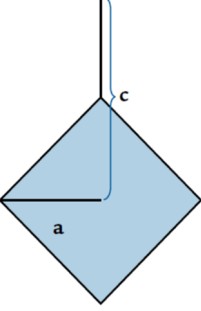

**Figure 5.** Schematic diagram of Vickers indenter and measurement of indentation cracking. C = crack length (from the center of indent), a = length of half diagonal, and C ≥ 2a [19].

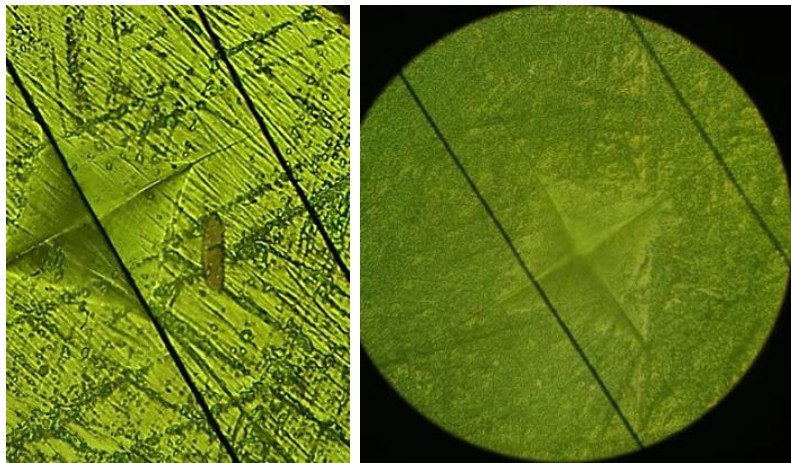

**Figure 6.** The tested sample under the microscope lens. On the left, C < a, so the load had to be increased. On the right, C ≥ a, so the load applied was accepted.

### 2.5. Brittleness Index

The brittleness index (B) of each sample was computed using the equation below:

$$B = VH/K,$$

where H is the hardness and K is the fracture toughness.

### 2.6. Statistical Analysis

Statistical software was used to examine the measured data (SPSS Statistics v25.0, Chicago, IL, USA). The normality of the distributions was analyzed and the groups were confirmed to be normally distributed. Data were evaluated using one-way analysis of variance (ANOVA) with multiple comparison corrections. The statistical significance level was chosen at $p < 0.05$.

## 3. Results

Table 4 shows descriptive data including means and standard deviations for each characteristic (flexural strength, hardness, and fracture toughness) of each material. Table 5 shows multiple comparisons among the values of the non-aged categories, and Table 6 shows values for the same materials, both aged and non-aged.

**Table 4.** Means and standard deviations for biaxial flexural strength (S), Vickers hardness (VH), fracture toughness (K), and brittleness (B) according to the experimental groups.

| | Aging | S (MPa) (n = 18) | VH (MPa) (n = 14) | K (MPa.m$^{1/2}$) (n = 14) | B (μm$^{-1/2}$) (n = 14) |
|---|---|---|---|---|---|
| GNX | without | 874.1 ± 151.50 | 1652.0 ± 95.52 | 2.6 ± 0.24 | 6.3 |
| | with | 943.3 ± 164.906 | 1565.6 ± 147.63 | 3.0 ± 0.13 | 5.2 |
| ZCP | without | 765.3 ± 96.09 | 1614.4 ± 173.39 | 2.1 ± 0.05 | 7.8 |
| | with | 852. ± 146.50 | 1618.2 ± 58.13 | 3.7 ± 0.067 | 4.4 |
| UPC | without | 699.0 ± 85.41 | 1575.7 ± 98.75 | 3.2 ± 0.21 | 4.9 |
| | with | 715.6 ± 91.32 | 1691.4 ± 70.18 | 2.9 ± 0.079 | 5.8 |
| ENM | without | 175.8 ± 17.21 | 290.1 ± 31.49 | 0.7 ± 0.03 | 4.5 |
| | with | 150.3 ± 16.61 | 268.3 ± 15.89 | 0.6 ± 0.01 | 4.7 |
| EMX | without | 433.0 ± 54.61 | 721.9 ± 36.05 | 0.9 ± 0.03 | 7.9 |
| | with | 475.5 ± 56.19 | 681.7 ± 64.50 | 1.0 ± 0.048 | 7.1 |

**Table 5.** Multiple comparisons between values without aging. Shaded cells are statistically significant ($p < 0.05$).

| (I) Material | (J) Material | S Value Sig. | VH Value Sig. | K Value Sig. |
|---|---|---|---|---|
| GNX | ZCP | 0.402 | 0.905 | 0.004 |
| | UPC | 0.064 | 0.100 | 0.003 |
| | ENM | 0.000 | 0.000 | 0.000 |
| | EMX | 0.000 | 0.000 | 0.000 |
| ZCP | UPC | 0.549 | 0.899 | 0.000 |
| | ENM | 0.000 | 0.000 | 0.000 |
| | EMX | 0.000 | 0.000 | 0.000 |
| UPC | ENM | 0.000 | 0.000 | 0.000 |
| | EMX | 0.000 | 0.000 | 0.000 |
| ENM | EMX | 0.000 | 0.000 | 0.000 |

**Table 6.** Comparison of values within the same material, with and without aging. Shaded cells are statistically significant ($p < 0.05$).

| Material | Aging | S Value $n = 9$ Sig. (2-Tailed) | VH Value $n = 7$ Sig. (2-Tailed) | K Value $n = 7$ Sig. (2-Tailed) |
|---|---|---|---|---|
| GNX | without | 0.368 | 0.030 | 0.005 |
| | with | 0.368 | 0.030 | 0.005 |
| ZCP | without | 0.158 | 0.925 | 0.000 |
| | with | 0.158 | 0.925 | 0.000 |
| UPC | without | 0.696 | 0.000 | 0.005 |
| | with | 0.696 | 0.000 | 0.005 |
| ENM | without | 0.006 | 0.008 | 0.000 |
| | with | 0.006 | 0.008 | 0.000 |
| EMX | without | 0.123 | 0.018 | 0.071 |
| | with | 0.123 | 0.018 | 0.071 |

When comparing S values at baseline between groups, the non-aged GNX, ZCP, and UPC groups showed a significantly higher S value compared to the ENM and EMX groups ($p < 0.05$). The ENM group showed a significantly lower S value compared to the EMX group ($p < 0.05$) and was the only material showing a significant reduction in S value with aging, as shown in Table 5.

When comparing Vickers hardness (VH) values at baseline between groups, the non-aged GNX, ZCP, and UPC groups showed a significantly higher VH value compared to the ENM and EMX groups ($p < 0.05$). The ENM group showed a significantly lower VH value compared to the EMX group ($p < 0.05$). With aging, GNX, ENM, and EMX had a reduced VH value ($p < 0.05$), while UPC increased its VH value with aging ($p < 0.05$).

When comparing fracture toughness (K) values at baseline between groups, the GNX group showed a significantly higher K value compared to the ZCP, ENM, and EMX groups ($p < 0.05$) and a significantly lower K value compared to UPC ($p < 0.05$). The ZCP group showed a significantly higher K value compared to the ENM and EMX groups ($p < 0.05$) and a significantly lower K value compared to the UPC group ($p < 0.05$). The ENM group showed a significantly lower K value compared to the EMX group ($p < 0.05$). With aging, the GNX and ZCP groups' K values improved, but those of the UPC and ENM groups declined ($p < 0.05$), as shown in Table 6.

## 4. Discussion

Mechanical property testing is a common way to evaluate dental materials which involves analyzing basic samples and loading them until failure. However, comparing

results can be challenging if testing conditions differ from prior research or manufacturer disclosures (e.g., sample sizes, shapes, or production sequences). To ensure accurate comparison, clear reporting and additional analysis may be necessary [20,21]. Depending on the purpose of the inquiry, samples could be obtained from any stratum of the restorative material, i.e., enamel, middle or dentin sides, or full thickness.

*4.1. Biaxial Flexural Strength (S)*

The mechanical testing machine for strength can use different methods such as uniaxial or biaxial bending tests to determine the strength of dental zirconia. These can use a three-point contact bending test (one upper and two lower point contacts for bar samples), a four-point contact bending test (two upper and two lower point contacts for bar samples), or a multiple-point contact bending test (one upper and multiple lower point contacts for disc samples) [21]. The biaxial flexure strength method is preferred as it applies even distribution of stresses resulting in multiple failure planes and better simulates the complex loading conditions in the mouth. This method is considered a more accurate measure of the mechanical strength of dental zirconia in clinical situations and has been added to the ISO standard for dental ceramics (ISO 6872-1994) [22].

Biaxial flexural strength tests can use three alternative designs for the upper indenter and the lower member, which are the ring indenter on a larger lower ring, the ball indenter on a lower ring, or the upper piston on the lower three balls. The Ball-on-Three-Balls-test (B3B) and the Ring-on-Ring-test (RoR) are both suitable for testing the strength of dental ceramics, but there are differences in the test procedures and potential sources of error. The B3B test is suitable for testing small or thin specimens but may overestimate strength if compliant interlayers are used. The RoR test mandates frictionless testing and uniform load introduction, and any deviations may cause overestimation or underestimation of the specimen strength. Compliant interlayers can help achieve uniform load introduction and reduce friction in the RoR test [23]. The B3B test is the ASTM standard for biaxial flexure testing and is more specific than the ISO standards [24]. In the study reported here, disc-shaped samples were chosen and placed in a housing to prevent lateral movement with the upper piston contacting the center of the sample on the lower three balls.

Different subtypes of esthetic dental materials require special attention, as even those with the same commercial name can have different mechanical strengths due to different ceramic mixtures and translucencies [25]. For example, IPS e.max® ZirCAD Prime and IPS e.max® ZirCAD Prime Esthetic seem to be similar; however, they have an incisal layer of 5Y-TZP/dentin layer of 3Y-TZP and an incisal layer of5Y-TZP/dentin layer of 4Y-TZP, respectively. Their strengths are 650 MPa (incisal)/1200 MPa (dentin) and 650 MPa (incisal)/850 MPa (dentin), respectively. Thus, they are clinically indicated for different prostheses (the first for crowns, three-unit bridges, and long bridges using three or four or more units with a maximum of two pontics. The second is for crowns and three-unit bridges only).

Some of these subtypes achieve different translucencies through various methods, such as altering the processing conditions or adding other materials to the ceramic matrix. The goal is to create a more homogeneous structure with fewer defects and a higher degree of crystallinity, which in turn leads to improved mechanical and optical properties. Adding new materials or pigments alters the shades [26,27]. Sintering parameters affect crystal structure size, material density, and phase shifts, which ultimately affect mechanical properties. Increasing the yttria in 5Y-TZP creates more isotropic phases, which reduces flexural strength and fracture toughness [24]. Multilayered translucent monolithic zirconia can mimic color gradients, but pigments are considered contaminants and can affect the microstructure, strength, hardness, and toughness [25–29]. The strength of each layer of these combinations could not be verified in this study since samples contained the whole disc thickness. The material Ceramill Zolid had four layers, IPS e.max® ZirCAD Prime had two, and Upcera had five layers, with the remaining two materials being single-layered.

As expected when inspecting the composition of the materials, the results of our study showed the higher fracture strength of the three zirconia monolithic materials over the polymer-infiltrated hybrid ceramic and the lithium disilicate materials. The hybrid ceramic had significantly lower values than all other materials.

Nevertheless, there were no studies to directly compare with for the first three materials, as they are new and no published data is present for them (search terms on PubMed: strength, Ceramill Zolid Gen-X, Upcera Explore Esthetic), as shown in Table 7. IPS e.max ZirCAD Prime had 10 citations; three were for flexural strength [30–32] and two were for bar-shaped samples using three- and four-point contact [31,32]. Winter et al. (2022) studied the effect of thermal aging on three zirconia materials [31]. Both ZirCAD and Optimill exhibited consecutive increases in their flexural strength. Layer four (the deepest, 3Y) of ZirCAD displayed the highest flexural strength both before and after artificial aging. The Weibull modulus varied between 4.32 (for ZirCAD layer one) and 13.58 (for Ceramill zolid fx multilayer layer four) after thermal cycling. ZirCAD exhibited the highest Vickers hardness overall, with layer one (enamel side) displaying the highest value (1579.18 ± 47.14 HV) before aging and layer two showing the highest value (1607.1 ± 149.71 HV) after aging. The flexural strength and Vickers hardness varied significantly across the four ZirCAD layers. The mechanical properties were not significantly affected by thermal aging.

**Table 7.** Studies reporting mechanical test values for materials used in this investigation.

| Material | Biaxial Flexure Strength | /Aging | Fracture Toughness | /Aging | Vickers Hardness | /Aging | Brittleness | /Aging |
|---|---|---|---|---|---|---|---|---|
| Ceramill Zolid Gen X | none | none | none | none | none | none | none | none |
| ZirCad prime | Multiple studies. | none | none | none | none | none | none | none |
| Upcera Explore Esthetic | none | none | none | none | none | none | none | none |
| Vita Enamic | Multiple studies | Multiple studies | Multiple studies | none | Multiple studies | none | none | none |
| IPS e.max CAD | Multiple studies | none | Multiple studies | none | Multiple studies | none | none | none |

The reported values from the manufacturer leaflets were generally higher than the results of this study. For example, the manufacturer of Ceramill Zolid has seven subtypes with the same name. No research reported the values regarding the subtype Ceramill Zolid Gen x, while the manufacturer reported a bending strength of 1000 +/− 150 MPa, but without mentioning the testing standards regarding specimen type, size, or indenter details. The Vickers hardness test result was reported by the manufacturer to be 1300 +/− 200 [33]. Upcera also has 11 subtypes beginning with the same name. The subtype Upcera Explore Esthetic had reported flexural strength (three-point bending test) values as: 1st layer ≥ 800 MPa, 2nd layer ≥ 850 MPa, 3rd layer ≥ 900 MPa, 4th layer ≥ 1000 MPa, and 5th layer ≥ 1100 MPa on its website, but, once more, nothing was reported as to which testing conditions were used [34]. For ZirCad prime, the manufacturer reports flexural strength values of 650 MPa (Incisal) and 1200 MPa (Dentin) [35], which are close to the results of this study.

Vita Enamic subtype 2M2 was reported in four papers, all related to optical properties. The general Vita Enamic term was tested for biaxial flexure strength in six studies [36–41] on biaxial fracture strength (ranging from 100.0 ± 3.2 MPa [36] to 174 ± 13 [39]) but none had reported the specific subtype used in them. There are six available search results [42–44] (search terms: strength, IPS Emax CAD) that tested flexural strength where samples were not bonded. Three were bar-shaped samples [43–45]. Oliveira Junior et al. [46] (2022) used CAD-CAM blocks that were milled into cylinders (Ø = 10.0 mm) and sliced into disks of approximately 1.3 mm ±0.02 mm with a precision saw. They found that exposure to simulated gastric juice and brushing resulted in significant changes in the physical prop-

erties of the CAD-CAM monolithic materials. The materials showed decreased hardness, increased substance loss, and decreased flexural strength after exposure. The value of flexural strength before exposure was reported to be $165.9 \pm 38.8$ for IPS Empress CAD and $146.8 \pm 14.1$ for Vita Enamic. Wang et al. (2019) [47] found that heat-pressed lithium disilicate glass ceramics (IPS Emax press) exhibited higher flexural strength than CAD/CAM equivalents (IPS Emax CAD), according to their findings. Heat-pressed crystals were longer and broader, whereas CAD/CAM crystals were shorter and wider. A larger crystal aspect ratio may increase fracture propagation resistance in glass-ceramics by enhancing the "interlocking effect" of the crystalline phase. Fabian Fonzar R et al., 2017 [48], found results for IPS e.max Press and CAD indicating that the flexural strength of the Press and CAD specimens did not differ considerably. However, varied translucencies within the Press group demonstrated comparable flexural strength. In contrast, there were statistically significant variations across the assessed translucencies within the CAD group. MT, in particular, demonstrated much better flexural strength than HT and MO. Furthermore, LT has substantially greater flexural strength than MO. The flexural strength values reported in this study for IPS Emax CAD ($433.0 \pm 54.61$) were higher than those reported by Kang et al. [49], Buso et al. [50], and Lin et al. [51] [mean flexural strength (SD) of 408.3 (855.9), 416.1 (50.1) MPa, and 365.1 (46.0) MPa].

To best imitate the oral environment, a combination of thermal and mechanical cycling has been advocated for [52]. Then specimens in this study were hydrothermally aged for 5 h at 134 °C and 2 bar (200 KPa) in the current investigation. This approach was chosen because one-hour autoclave aging at 134 °C and 2 bar (200 KPa) corresponds to three to four years of clinical usage and causes significant tetragonal-monoclinic phase transition (about 55–80% monoclinic phase content) [13]. Responding to criticism that this test does not equate to actual clinical situations and may overestimate the real deterioration of zirconia ceramics in the oral cavity, we contend that it is still a valuable tool for estimating these materials' long-term durability and comparing between different materials subjected to the same conditions [13]. The response of dental zirconia materials to hydrothermal aging can vary depending on their yttria content. Two studies [52,53] by Kocjan and colleagues investigated the in vivo aging of different zirconia materials: a 3Y-TZP, a 4Y-TZP, and a 5Y-TZP. In their first study, they found that all three materials experienced a significant decrease in flexural strength and an increase in surface roughness after a 5-year equivalent aging process, with 5Y-TZP showing the least amount of strength loss. In their second study, they found that all three materials showed signs of phase transformation and microstructure changes after aging, with 3Y-TZP showing the greatest amount of monoclinic phase formation and color change. The 3Y zirconia has the lowest yttria content and the highest susceptibility to aging-related degradation, such as low-temperature degradation (LTD). The 4Y zirconia has a slightly higher yttria content than 3Y and is more stable and resistant to aging-related degradation but may have limited potential for transformation toughening due to its lower tetragonal content. The 5Y zirconia has the highest yttria content and is the most stable and resistant to aging-related degradation but may also have limited potential for transformation toughening. The difference in zirconia materials and their response to aging is complex and depends on various factors such as composition, microstructure, processing conditions, surface treatments, and aging conditions [54]. Kim et al. [55] analyzed the effects of aging in a steam autoclave on the monoclinic (m) zirconia content of various surface treatments of zirconia specimens. The study found that before aging, the as-received surfaces had negligible amounts of m-phase, while grit-blasted and ground surfaces had around 5% m-phase. After 2 h of aging, grit-blasted surfaces showed around 12% m-phase, while ground surfaces exhibited around 7–8% m-phase. As aging time accumulated, the amount of m-phase increased steadily in both grit-blasted and ground surfaces. The as-received surfaces initially had the lowest amount of m-phase, but caught up with the ground surfaces after 10 h of aging. After 20 h of aging, the CAD/CAM-machined control surfaces had over 55% m-phase, while grit-blasted surfaces had around

30%, 80- and 120-grit ground surfaces had around 20%, and 600-grit ground surfaces had around 15% m-phase.

In our study, the surface roughness produced later as a result of LTD could not be evaluated due to the lack of occlusal force simulations or friction which are expected to be causing the dislodgement of surface grains, thus increasing roughness. With aging, the tested materials maintained the same rankings, with Enamic hybrid ceramic being the most affected by the aging process (175.8 to 150.3 MPa). Lucsanszky and Ruse [56] showed the same result, as the flexural strength of Enamic was significantly decreased with aging in 37 °C distilled water for 30 days from 148.16 to 135 MPa, despite using different protocols for aging. In addition, Ferhan Egilmez et al. [57] showed that water storage (37 °C for 3 weeks), autoclave treatment (134 °C at 200 kPa for 12 h), and thermal cycling (5000 times at 5–55 °C) significantly decreased the flexural strength of Enamic. These results could be explained by the hydrolysis of the matrix-filler interface combined with water penetration in the matrix, causing the network to soften and swell and reducing frictional forces between polymer chains.

### 4.2. Hardness Evaluation

When comparing hardness values at baseline between groups, the GNX, ZCP, and UPC groups showed a significantly higher VH value compared to that of the ENM and EMX groups ($p < 0.050$). The ENM group showed a significantly lower VH value compared to the EMX group ($p < 0.05$). GNX had the highest values without aging, UPC had the highest values with aging, and ENM had the lowest VH with and without aging.

This discrepancy was thought to be due to differences in the components and structuring of these materials. Candido et al. [58] demonstrated that the Vickers hardness of monolithic zirconia (Prettau Zircon) was 1452.16, which is around the average of the monolithic zirconia determined in this research, with GNX 1652.0344 > ZCP 1614.3894 > UPC 1575.6723 (without aging). Even after aging, zirconia has much greater hardness readings than those with polymer resin and glass components.

According to the literature, hardness values of enamel and dentin were 2–3.5 GPa and 0.3–0.7 GPa, respectively [59]. Zirconia is significantly harder (10–12 GPa) than enamel (<6 GPa) [60]. In this research, Vita Enamic seemed to have a hardness five times lower than zirconia yet is similar to human enamel, which is consistent with the findings of other authors [61]. As per Kim et al. [62], ENM and EMX (290.0667 and 721.8804) showed a considerable decline in hardness with age (268.3171 and 681.7150). Despite having a superior wear impact on the antagonist tooth, ENM does have greater wear compared to ceramic-based materials [63].

The hardness response of dental zirconia materials to hydrothermal aging can vary, depending on factors such as the yttria content of the material. Some studies have reported a decrease in hardness after aging [64], while others have shown an increase or no significant change [65–67]. For 3Y-LT and 4Y-HT, surface and bulk properties were affected by aging to a similar extent. However, surface and bulk properties may change during clinical use as a result of prolonged degradation of Y-SZ, and the response may also vary depending on the testing method used.

Gaillard et al. [68] observed that hydrothermal deterioration at 131 °C in water vapor for 1–60 h induced t–m phase transformation and propagation of cracks under the surface, both of which were related to a reduction in hardness, which partially concurred with this study because GNX and ZCP showed a significant and insignificant reduction in hardness, respectively, whereas UPC showed a significant increase. Higher UPC hardness numbers may be correlated with a low quantity of tetragonal to monoclinic (t–m) transition [69]. When a restricted t–m transition is localized superficially, the material's hardness may rise.

### 4.3. Fracture Toughness Evaluation (K)

When fracture toughness values (K) from baseline were compared between groups, the GNX group had a considerably higher K value than the ZCP, ENM, and EMX groups

($p < 0.05$), and a substantially lower K value than the UPC group ($p < 0.05$). The ENM group had a notably lower K value than the EMX group ($p < 0.05$). The discrepancy was thought to be due to differences in the components and microstructures of these materials. The mechanical characteristics of Y-TZP are affected by particle size. At a certain size, the stability of Y-TZP declines, and it becomes more susceptible to t–m transformation. The transformation ratio reduces in the presence of smaller particles (<1 μm). Furthermore, transformation is unattainable below a particular particle size (~0.2 μm), resulting in a loss of fracture toughness [10].

According to the literature [59], fracture toughness values of enamel and dentin were 0.67–3.93 MPa·m$^{1/2}$ and 1.1–2.3 MPa·m$^{1/2}$, respectively. In this study, fracture toughness values ranged from 2.0781 to 3.2271 for highly translucent zirconia and 0.6500 for Enamic, which is on par with the previous literature [59,70]. EMX had a K mean value of 0.9193, which is slightly less than the literature, which indicated 1.28 ± 0.19 [71].

The response of dental zirconia materials to fracture toughness after hydrothermal aging can vary. Studies have shown that the fracture toughness of some zirconia materials may decrease after aging, while others may remain relatively stable. According to prior research, 5Y-PSZ demonstrated both strong aging resistance and maintained translucency stability [72]. On the other hand, 3Y-TZP experienced a notable reduction in characteristic stress but showed an increase in fracture toughness following the aging process. The variation in response can be influenced by factors such as the yttria content, sintering temperature, aging conditions, and material of infiltration [73,74]. Infiltration of 3Y-TZP demonstrated better resistance to change after aging. Studies investigated the effects of co-doping silica, alumina, and lanthanum in zirconia to improve aging resistance. Co-doping with alumina and lanthanum resulted in better aging resistance by changing the relationship between nucleation and growth and creating strongly bonded oxygen vacancies that slow down diffusion of water molecules at the grain boundary. Co-doping with silica and alumina also showed improved aging resistance due to strong grain boundaries and rounded glassy grains reducing internal stresses. Mechanical properties were not significantly affected in graded zirconia but showed reduced flexural strength in nongraded species. Similar results were found for fracture toughness in silica-alumina co-doped zirconia [74].

Without aging, UPC had the highest toughness values, whereas ZCP had the highest values with aging. With and without aging, ENM had the lowest K. Without aging, GNX and ZCP had considerably lower K values than with aging ($p < 0.05$). This rise in K values may be related to the tetragonal to monoclinic (t–m) transition [69]. When UPC and ENM were not aged, they had considerably higher K values ($p < 0.05$). Zhang et al. [75] made several intriguing observations on the toughening processes of the investigated materials. 4Y-TZP had a lower toughening impact than 3Y-TZP, owing to an increase in the proportion of non-transformable cubic phase as well as reduced tetragonality of the residual tetragonal phase. Nevertheless, there was no indication of transition toughening in 5Y-TZP. This was not the case in this investigation, since GNX (4Y-TZP) had a greater K than ZCP (3Y-TZP and 5Y-TZP), and yet a lower K than UPC (4Y-TZP and 5Y-TZP). According to this study, tetragonal zirconia with a greater yttria concentration is more stable, with less ability for transition toughening and hence a reduced low-temperature deterioration (LTD) tendency [76]. The GNX (4Y-TZP) has a yttria concentration of up to 6–7%, and ZCP (3Y-TZP and 5Y-TZP) has a concentration of 4.5–7%, showing significantly increased K values with aging, whereas UPC (4Y-TZP and 5Y-TZP) has a yttria concentration of 5.8–9.7% and a significantly decreased K value with aging [77].

Polymer-infiltrated ceramics limit crack propagation via polymer interpenetration in the structure [62]. Lucsanszky and Ruse [56] showed that the fracture toughness of Enamic was significantly increased with aging in 37 °C distilled water for 30 days from 0.83 to 1 MPa.m$^{1/2}$, which disagrees with this study, as the K value significantly decreased with aging for ENM. Similarly to this study, Hampe et al. [78] showed a decrease in ENM's K value, but it was insignificant. EMX without aging showed an insignificantly lower K value

compared to that shown with aging ($p > 0.05$). This is supported by Hampe's findings. Tangential compressive stresses are found on the sides of the crystals because of a disparity in thermal expansion between the glass matrix and the crystals. Increased strength, S, and fracture toughness, K, are caused by these stresses [78].

*4.4. Brittleness Evaluation*

Brittleness index (B) is a measure of a material's relative sensitivity to deformation and fracture that is claimed to be directly connected to dental ceramic machinability. It may be employed as a scale in the development of the materials used for crown replacement, as well as a quantifiable measure of mechanical characteristic deterioration.

According to the literature, the brittleness index range for dental restorative glasses and ceramics should be kept around 3–9 $\mu m^{-1/2}$ [79], which is consistent with the findings in this investigation. There are no previous studies to compare with for any of our materials, except for IPS e.max CAD, where the following values were reported: 2.4 [80], 2.72 (0.17) [81], 1.77 (0.28) [82], and 2.90 (0.54) [83]. All these values are lower than what was found here (7.852). No studies reported on the effect of autoclave aging on the values of the brittleness index.

The following is the order of materials based on their brittleness, without aging:

EMX 7.852 > ZCP 7.768 > GNX 6.260 > UPC 4.882 > ENM 4.463.

With aging, GNX and ZCP significantly reduced in value, changing the order to the following:

EMX 7.100 > UPC 5.827 > GNX 5.230 > ENM 4.653 > ZCP 4.369.

*4.5. Limitations and Consideration for Further Studies*

Although this provides essential data for dental clinicians, the researchers of this study acknowledge that this study only tested for mechanical property variables, not including all clinical or laboratory steps affecting these materials in vivo. Furthermore, samples were not designed to be full-contoured restorations and lacked the presence of other relevant materials in contact with the samples, such as dental cements and natural dental tissue. Surface changes, such as micro-roughness, were not studied, nor were the effects of different fatigues.

More research, both in vivo and in vitro, is required to clearly establish clinical indications, physical-mechanical properties, constraints, and protracted performance of these kinds of restorations.

**5. Conclusions**

Based on the results obtained and within the limitations of this study, the following conclusions can be derived:

1. At baseline, monolithic zirconia showed higher mechanical properties (flexural strength, hardness, and fracture toughness) compared with lithium disilicate, while hybrid ceramic ENM showed the lowest mechanical properties. However, all materials meet clinically accepted values for their indications.
2. Monolithic zirconia groups showed a lower brittleness index compared with lithium disilicate, while hybrid ceramic ENM showed the lowest brittleness index. Consequently, ENM monolithic zirconia may have better machinability.
3. All groups showed, to some extent, changes in mechanical properties (flexural strength, hardness, fracture toughness, and brittleness) with aging which was within the clinical acceptability range.

**Author Contributions:** Conceptualization, L.A.A.-N. and S.N.A.; Formal analysis, B.A.; Funding acquisition, B.A.; Investigation, L.A.A.-N., B.A. and S.N.A.; Methodology, L.A.A.-N., S.N.A., and B.A.; Project administration, L.A.A.-N. and S.N.A.; Supervision, L.A.A.-N. and S.N.A.; Writing—original draft, L.A.A.-N., S.N.A. and B.A.; Writing—review & editing, S.N.A. and L.A.A.-N. All authors have read and agreed to the published version of the manuscript.

**Funding:** This research was funded by the Faculty of Scientific Research (grant number 20210249-7984) at Jordan University of Science and Technology.

**Institutional Review Board Statement:** Not applicable.

**Informed Consent Statement:** Not applicable.

**Data Availability Statement:** Not applicable.

**Conflicts of Interest:** The authors declare no conflict of interest. The funders had no role in the design of the study; in the collection, analyses, or interpretation of data; in the writing of the manuscript, or in the decision to publish the results.

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
