# Peer review of "Mechanical Properties of Five Esthetic Ceramic Materials Used for Monolithic Restorations: A Comparative In Vitro Study"

_ceramics, doi:10.3390/ceramics6020061_

Round 1
Reviewer 1 Report
In this manuscript the authors compare mechanical properties of 3 different dental zirconia types, one glass ceramic and a hybrid material. They investigate biaxial flexural strength, toughness, hardness and brittleness without and with aging. The experimental design is quite straight forward and the results are to be expected, but there are many issues with this manuscript, that need to be addressed before it can be published.
Broad comments:
1. Language, spelling, grammar is fine, but needs some minor checking.
2. The general idea of the study is strange. They compare 3 different types of material, 2 of which are ceramics, and one is a hybrid material = ceramic-polymer composite. Out of the 2 ceramic material types one is monolithic zirconia, and the other is lithium disilicate glass ceramic. Of course these 3 types of materials will vary widely in mechanical strength, which can also be seen from the indications! There is no surprise here, that zirconia is much stronger then glass ceramic, which in turn is much stronger then the hybrid “ceramic”. This was all to be expected, so why did you investigate it? The aim needs to be more specified.
3. The authors keep saying, that there are almost no studies regarding these materials. But it seems like the references were collected a few years ago, as most of them are quite old. Only 12 out of 66 references are from 2020 or more recent. As the zirconia multilayer materials and zirconia materials with higher yttria content are a fairly recent development in dental applications, there are less studies before 2020. The authors need to update the references!
4. There seems to be a general lack of understanding regarding the influence of composition on mechanical strength. Especially in zirconia materials, where yttria content strongly influences the mechanical strength and translucency. Lower yttria content (3 mol%) means high strength and low translucency. Increased yttria content leads to up to 50 % strength reduction with an increase in translucency. This is of course related to microstructure! A good reference is citation [33] – Stawaczyk et al, which they cited but don’t seem to understand well.
5. Many of the figures are of the level of a student work. Fig. 1, 2, 5, 6, 7, 8 offer no value to this study. Images of equipment should be exchanged with schematics of the experiment (such as biaxial flexural strength set-up or hardness testing), if this is considered necessary.
6. Roughness values should be added after sintering and polishing, especially for the zirconia materials before and after aging!
Specific comments:
- Abstract, p. 1, l. 23: “, and fully crystallized” – Zirconia blocks are sintered, Emax CAD is crystallized, as it consists of a glassy material for milling, and Vita Enamic should not be crystallized or temperature treated whatsoever. Which the authors know, as they describe differently in the text.
- Introduction, p.1-2, l. 42 -46: needs to be updated about a) different types and generations of zirconia today and b) advances in multilayer ceramics. The cited reference is from 2014, the focus has very much shifted from influence of Ce and Mg, most of the dental zirconia materials are stabilized with various amounts of yttria!
- Reference 9 is only a link (please check how to correctly refer to online sources), Ref. 10 contains nothing!
- Mat. & methods, p. 3, l.104: please a paragraph between text and table caption.
- Mat & methods, p.3, table 1: what is the unit of the composition? Wt.% or mol.%? Please add
- Mat & methods, p.4, fig. 1: does this mean that samples were cut upright out of the disc? This would have an influence on the multilayer material ZirCAD prime, as the incisal part is 5Y and the body part of the disc is 3Y. Meaning you have a material gradient in each sample. This surely has an influence on all of the values you determined! This needs to be discussed! Especially for hardness and K-values, as they are determined locally, so if you put the indenter into a 5Y area, this might give a different value then in the 3Y area.
- P.4, l.128: “table 1. X/250, Devco S.r.l. Italy)” what does this mean?
- P.5, Fig. 3: you polished the zirconia materials before sintering? Why? The material is like sand or gypsum. This seems very wrong.
- P.5, l. 137: “Zirconia specimens.. were completely crystallized..” – please change to sintered! The presintered discs contain already fully crystallized particles.
- P. 5, tab.2, l. 144: ENM should under no circumstances be sintered, as it contains a polymer matrix, which would burn in a sinter process! Please update!
- P. 7, VH: the indenter used in Vickers hardness has the shape of a 4-sided pyramid, so the indent usually has a square shape. There you measure the diagonals, not diameters (they only apply to circles!) Please rephrase!
- P.8, L210 ff: please format equations correctly.
- P. 10, table 4: please round all values to one digit after the decimal point in case of S, VH values and 2 digits in case of K and B values! Meaning: S 874.1 MPa is sufficient.
- P. 11, Discussion: please please update the references! First section of the discussion – reference 20 is from 2017! So many studies and developments since then!
- P. Jevnikar and colleagues investigate in vivo aging of different zirconia materials in the oral cavity in 2020: part i: https://doi.org/10.1016/j.dental.2020.11.023, part ii: https://doi.org/10.1016/j.dental.2020.11.019, others also looked into this!
- The discussion is very bloated and overloaded with unnecessary information. L. 277 – 292 can be strongly reduced, as you did not investigate the influence of sample preparation and design.
- Table 7 and 8 are not necessary and should be deleted.
- Table 9: please give the references here.
- Table 9: It seems futile to look for the exact materials used in this study. There are many different producers for dental zirconia, and each (as already described by the authors) offers a large variety of zirconia materials, with different shapes, compositions, shades, possible multilayers, and so on. To expect to find the exact same materials studied is strange. Rather the overlying characteristic differences (e.g. yttria content, alumina content, material or shade multilayer) should be discussed and used for comparison.
- P.13, l.309-323: section can be deleted
- P.13, l. 341: “size of growth” – sinter parameters influence the grain growth, and therefor the grain size, but not the size of growth. Please rephrase.
- P.14, discussion about Flex strength S: authors say that the companies don’t give information about how the strength values were measured. If you look into the data sheet of ZirCAD prime and Zolid gen x, you will find that they measured in 3 point bending according to the ISO standard 6872. I am sure the upcera will offer similar information if asked.
- This part of the discussion should focus on the influence of the composition of the different materials on the flex strength. GNX is a 4Y material, ZCP is 3Y in body and 5Y in the incisal area. UPC is not clear, EMX is a glass ceramic and ENM is a composite material. Here the differences in the zirconia materials and the influence of the aging should be discussed. Instead the authors focus on the perceived lack of studies on the exact same materials. 3Y has a higher tetragonal content, which will lead to so called transformation toughening. This will not happen in monoclinic content, which is more prevalent in higher Y content ceramics. But they found that GNX (4Y) and ZCP (3Y/5Y) both increased S after aging. So what is happening there? Same with EMX? Why?
-
Author Response
Thanks a lot for your effort and valuable comments. Please find attached the responses to your comments.

Reviewer 2 Report
I think the article "Mechanical Properties of Five Esthetic Ceramic Materials Used for Monolithic Restorations: A Comparative In Vitro Study" is very well structured and comprehensive. The introduction provides interesting and suitable background. The study results are very interesting and are clearly presented. The authors' conclusions are well supported by the experimental results.
Author Response
Thanks a lot for your effort and valuable comments. We are so grateful for your positive and encouraging feedback. Highly appreciated.
Reviewer 3 Report
Dear authors, first of all congratulations for your work.
In order to enhance the quality of your scientific work I suggest to improve the Introduction, which, in my opinion is rather short and it contains a too small number of articles.
I suggest some:
Goujat, A., Abouelleil, H., Colon, P., Jeannin, C., Pradelle, N., Seux, D., & Grosgogeat, B. (2018). Mechanical properties and internal fit of 4 CAD-CAM block materials. The Journal of Prosthetic Dentistry, 119(3), 384–389. https://doi.org/https://doi.org/10.1016/j.prosdent.2017.03.001
https://ss.bjmu.edu.cn/Sites/Uploaded/File/2022/02/186378079173367470948765544.pdf
I appreciate as very good the Materials and Method Part and also the Results.
You should also revise the style of the References should follow the Instructions for authors. Reference number 9 and 10 are not clear. Please verify them.
Author Response

(The authors gave the same response as above.)

Round 2
Reviewer 1 Report
Dear authors,
thank you for revising the manuscript, the quality has improved. But there are still some minor issues, which should be updated before accepting this manuscript for publication:
- p.4, l. 150 - 155: "Polishing after (sintering) would produce more surface scratches and heat changes." - please give a reference, that has shown this. According to the ISO standard 6872 samples should be subjected to polishing after sintering to make sure that the sample surfaces are plane parallel.
- p. 11, l. 303: "or (one upper and..) - please add "biaxial flexural strength" or something like that before the bracket.
- p. 12, l. 351: "using 3 ad 4" - please change to "and"
- p. 12, l. 352 "Winter et al (2022)studied the effect of thermal thermal aging fo three zircoia material[31]." this sentence contains at least 4 mistakes.
- p. 12, l. 353: "Both ZirCAD and Optimill." - please combine with the next sentence
- p. 12, l. 364-5: "while the manufacturer reported a bending strength of 1000 +/- 150 MPa, but not mentioning the type of test nor which test standards were followed." - according to the main catalogue from Amann Girrbach, p. 107 Zolid Gen-x was tested by 3-point flexural strength [MPa] DIN EN ISO 6872. Same catalogue, p. 109: Flexural strength (4-point) 900 +/- 150 MPa.
- p. 12, l. 370f: "For ZirCad prime, the manufacturer reports a value of 850–1200 for brittleness ad are 650 MPa (Incisal) 1200 MPa (Dentin) for the flexural strength." What do you mean with brittleness? Please give references for these discussion points. Even documents supplied by the manufacturer need to be cited.
- p. 13, l. 376 - 378: please check the following references for flexural strength data of IPS Emax CAD: Kang, S. H., Chang, J., & Son, H. H. (2013). Flexural strength and microstructure of two lithium disilicate glass ceramics for CAD/CAM restoration in the dental clinic. Restorative dentistry & endodontics, 38(3), 134-140.; Wang, F., Yu, T., & Chen, J. (2019). Biaxial flexural strength and translucent characteristics of dental lithium disilicate glass ceramics with different translucencies. Journal of prosthodontic research, 64(1), 71-77.; Fonzar, R. F., Carrabba, M., Sedda, M., Ferrari, M., Goracci, C., & Vichi, A. (2017). Flexural resistance of heat-pressed and CAD-CAM lithium disilicate with different translucencies. Dental Materials, 33(1), 63-70.
Author Response
- p.4, l. 150 - 155: "Polishing after (sintering) would produce more surface scratches and heat changes." - please give a reference, that has shown this. According to the ISO standard 6872 samples should be subjected to polishing after sintering to make sure that the sample surfaces are plane parallel.
Thanks for the comment. A reference was provided and the order of the references was changed accordingly
- p. 11, l. 303: "or (one upper and..) - please add "biaxial flexural strength" or something like that before the bracket.
Thanks for the comment. Suitable words were added to the paragraph.
- p. 12, l. 351: "using 3 ad 4" - please change to "and"
Thanks for the comment. The word was corrected as requested.
- p. 12, l. 352 "Winter et al (2022)studied the effect of thermal thermal aging fo three zircoia material[31]." this sentence contains at least 4 mistakes.
Thanks for the comment. All mistakes were corrected.
- p. 12, l. 353: "Both ZirCAD and Optimill." - please combine with the next sentence
Thanks for the comment. The phrase was combined as requested.
- p. 12, l. 364-5: "while the manufacturer reported a bending strength of 1000 +/- 150 MPa, but not mentioning the type of test nor which test standards were followed." - according to the main catalogue from Amann Girrbach, p. 107 Zolid Gen-x was tested by 3-point flexural strength [MPa] DIN EN ISO 6872. Same catalogue, p. 109: Flexural strength (4-point) 900 +/- 150 MPa.
Thanks for the comment. The context was corrected in a way to refer to the intended goal as the manufacturing company has not mentioned the specimen shape, size, or the indenter details. Only the number of "points" has been mentioned, which does not reflect the exact testing standards.
- p. 12, l. 370f: "For ZirCad prime, the manufacturer reports a value of 850–1200 for brittleness ad are 650 MPa (Incisal) 1200 MPa (Dentin) for the flexural strength." What do you mean with brittleness? Please give references for these discussion points. Even documents supplied by the manufacturer need to be cited.
Thanks for the comment. The paragraph was rephrased and a reference was added.
- p. 13, l. 376 - 378: please check the following references for flexural strength data of IPS Emax CAD: Kang, S. H., Chang, J., & Son, H. H. (2013). Flexural strength and microstructure of two lithium disilicate glass ceramics for CAD/CAM restoration in the dental clinic. Restorative dentistry & endodontics, 38(3), 134-140.; Wang, F., Yu, T., & Chen, J. (2019). Biaxial flexural strength and translucent characteristics of dental lithium disilicate glass ceramics with different translucencies. Journal of prosthodontic research, 64(1), 71-77.; Fonzar, R. F., Carrabba, M., Sedda, M., Ferrari, M., Goracci, C., & Vichi, A. (2017). Flexural resistance of heat-pressed and CAD-CAM lithium disilicate with different translucencies. Dental Materials, 33(1), 63-70.
Thanks for the comment. They were double-checked.
